# Application of Bifurcated Semitendinosus Muscle Transposition for Treatment of Fecal Incontinence in Two Dogs

**DOI:** 10.3390/vetsci10020150

**Published:** 2023-02-13

**Authors:** Mu-Young Kim, Chang-Hoon Nam, Ji-Hyun Kim, Hun-Young Yoon

**Affiliations:** 1Department of Veterinary Clinical Sciences, College of Veterinary Medicine, Purdue University, West Lafayette, IN 47907, USA; 2Department of Veterinary Surgery, College of Veterinary Medicine, Konkuk University, 120 Neungdong-ro, Gwangjin-gu, Seoul 05029, Republic of Korea; 3VIP Animal Medical Center, Seoul 05029, Republic of Korea; 4KU Center for Animal Blood Medical Science, Konkuk University, Seoul 05029, Republic of Korea

**Keywords:** anal mass, augmentation, anal sphincter, dog, fecal incontinence, muscle, pull through, squamous cell carcinoma, transposition

## Abstract

**Simple Summary:**

Fecal incontinence is a condition characterized by an inability to control defecation, due to either a failure of the large intestine to contain colorectal contents or a failure of the anal sphincter to resist the rectal propulsive forces. Surgical intervention is often necessary to treat severe cases. This study presents a modified semitendinosus muscle flap technique for anal sphincter augmentation in two canine patients with fecal incontinence: a 4-year-old mixed breed dog and a 19-year-old English Cocker Spaniel. The semitendinosus muscle was bifurcated and transposed around the anus. The resulting two muscle bundles completely encircled the anorectum, leading to a complete reconstruction of the external anal sphincter. Both dogs regained fecal continence following surgery. This new technique requires a shorter length of harvested muscle flap and is a relatively simple surgical procedure compared to conventional surgeries. So, it may be a viable surgical option for anal sphincter incontinence in dogs.

**Abstract:**

A 4-year-old mixed breed dog and a 19-year-old English cocker spaniel dog were evaluated for fecal incontinence. The second dog’s fecal incontinence was associated with the anal mass. In both dogs, reconstruction of the external anal sphincter was required to gain fecal continence. Especially in the dog with an anal mass, the whole musculature involved in fecal continence was removed with the affected anorectum. Conventional surgical treatments for fecal incontinence have limitations in terms of muscle flap length and complexity of the surgical procedure. A modified surgical technique using the semitendinosus muscle was devised in the present study to overcome these limitations. The distal part of the semitendinosus muscle was bifurcated to make two muscle bundles, used to completely encircle the anorectum. These muscle bundles were sutured to the surrounding rectal muscle and the pelvic diaphragm to simulate the function of the external anal sphincter. Three months after surgery, both dogs showed significantly improved fecal continence without severe complications, such as infection, dehiscence, or lameness of the limb where the semitendinosus muscle was harvested. The outcomes of the two dogs supported the acceptability of the bifurcated muscle flap for anal sphincter augmentation. In addition, this report showed the possibility of more diverse applications of semitendinosus muscle in dogs.

## 1. Introduction

In animals, the process of fecal storage and evacuation depends on the maintenance of reservoir function and anal sphincter control, which requires complex coordination of the muscular and nervous systems [1,2,3]. The main muscular components in fecal continence include the internal and external anal sphincter, levator ani, coccygeus, and rectococcygeus muscles. Components of the nervous system include the pelvic, hypogastric, and motoric pudendal nerves [2]. Incoordination or dysfunction of these components results in fecal incontinence, defined as the inability to control defecation or involuntary loss of feces [4].

The main goal of medical therapy is to manage the underlying cause of fecal incontinence. Symptomatic medical therapy involves slowing transit time, reducing water content and bulk of feces, and increasing anal sphincter tone. However, in dogs, medical management alone is commonly insufficient to achieve fecal continence. In such cases, surgical treatment is indicated to augment or replace the anal sphincter function.

Various sphincter-enhancing procedures have been reported in dogs as myoplasty, fascial sling, and silicone elastomer [5,6,7,8,9,10]. The semitendinosus and levator ani muscle flap transposition techniques, used for myoplasty, successfully treated dogs with fecal incontinence caused by partial external anal sphincter defects [6,7]. Transposition of the sartorius muscle with electrical stimulation training, which creates a dynamic neosphincter, has been reported to increase the acute retention times of rectal contents in experimental dog models [11]. Transplantation of a latissimus dorsi muscle, with coaptation to the pudendal nerve and vessel anastomosis, was experimentally proven to restore voluntary anal continence in dogs [10]. Sato et al. demonstrated the potential use of the pudendal nerve to create a new reinforcing anal sphincter in dogs [9]. These two studies showed the possibility of utilizing the nerve anastomosis technique for canine fecal incontinence. Implantation of a fascial sling harvested from the tensor fasciae latae and silicone elastomer procedures were also proven to successfully correct fecal incontinence in dogs by increasing the passive intraluminal pressure of the rectum. The optimal surgical method is determined according to the cause and the defect area [5,6,10].

However, except for these studies, few clinical data are available on surgical treatment of fecal incontinence. The present study proposes a modified surgical technique to address the insufficiency of clinical evidence and the limitations of existing methods. This technique is based on conventional semitendinosus muscle flap transposition used for partial anal sphincter defects to treat total anal sphincter defects.

## 2. Case descriptions

### 2.1. Case 1

A 4-year-old spayed female mixed breed dog weighing 3.8 kg underwent rectovaginal fistula repair surgery two years ago, followed by repetitive revision surgery for dehiscence and infection. After one year of treatment, the symptoms related to rectovaginal fistula were resolved. However, fecal incontinence occurred while being treated for surgical complications. The referring veterinarian monitored the dog for one more year, with no improvement. The dog was referred to the Konkuk University Animal Medical Center (KUAMC) for surgical treatment.

On physical examination, the perineum was asymmetric, leading to the external anal orifice located slightly right of its normal position. The volume of muscular tissue surrounding the anorectal region was generally decreased. However, the skin and soft tissue on the right side of the anus were significantly more depressed than those on the left side. Perineal reflexes were reduced bilaterally, but the right side was weaker than the left. The anal orifice was widely open, and the rectal mucosa was exposed but not prolapsed (Figure 1). Standard laboratory blood tests were within normal ranges. The ultrasonographic findings revealed the absence of both anal sacs. The dog was diagnosed with anal sphincter incompetence, which led to fecal incontinence.

Anal sphincter enhancement surgery was performed using a semitendinosus muscle flap (Figure 2). The dog was positioned in sternal recumbency with the tail secured over the back. The cranial aspect of the hindlimbs was placed over the padded edge of the surgery table.

Bilateral skin incisions were made 1 cm away from the anus, and the incision on the right side was extended along the caudal margin of the right hind limb to the level of the distal one-third of the semitendinosus muscle (Figure 2A). Two tunnels between the left and right incisions were created at the dorsal and ventral sides of the anus (Figure 2B). After clearing the subcutaneous tissue, the muscle flap was carefully dissected from the surrounding tissue to preserve the integrity of the vasculature and nerve innervation, supplying the proximal part of the semitendinosus muscle. The semitendinosus muscle was transected at the level located between the middle and the distal one-third of it. The proximal attachment of the semitendinosus to the ischium remained intact. To entirely encircle the anus, the whole length of the semitendinosus muscle, from the origin to insertion points, needed to be transposed. To minimize the length of the muscle flap, a Y-shaped flap was made by longitudinal bifurcation of the distal half of the flap, dividing it into two bundles. The entire anus was encircled with a flap by passing each bundle through previously created tunnels. The flap was secured to the parts of the rectal muscle and pelvic diaphragm with simple interrupted sutures (3-0 polydioxanone suture) to simulate the external anal sphincter. To increase compression intensity produced by the muscle flap and to narrow the anal orifice, crotch sutures were placed between the bundle ends. Subcutaneous tissue and skin were closed in a routine manner. The dog recovered without any complication.

Four days after surgery, the dog started to show voluntary defecation, followed by involuntary passage of stool from the anus. The amount of inadvertent stool loss was significantly reduced over time. At the postoperative three-month follow-up, no signs of muscle flap disuse atrophy or avascular necrosis were observed, including skin depression around the anus, pain, inflammation, etc. Compared to the preoperative status, the opening of the anal orifice was significantly narrowed, and the rectal mucosa was less exposed. In addition, the dog gained almost normal defecation. However, a small amount of inadvertent stool occasionally continued to leak after voluntary defecation.

### 2.2. Case 2

A 19-year-old spayed female English cocker spaniel dog weighing 8.7 kg was referred for evaluation of chronic inflammation of the anus, anal mass, and constipation. This anal disorder had been treated with oral medications (laxatives, antibiotics) and topical corticosteroids; however, for the last three weeks, these treatments had no effects on constipation. As the inflammation worsened and the mass became larger, the dog struggled more with defecation. The owners reported that, for the last 5 days, the dog showed severe constipation and became completely anorexic. On initial physical examination, the dog was depressed and had a body condition score of 3/9. The anus was covered with solid, ulcerative, and hyperplastic tissue (Figure 3A). The little finger could not be passed through the anus due to the severely narrowed and stenotic anorectal canal. While examinations were carried out, the dog continued to try to defecate but failed. Complete blood count and serum biochemistry tests revealed leukocytosis (16.86 K/μL; reference range, 5.05–16.76 K/μL), low hematocrit levels (32.2%; reference range, 37.3–61.7%), hypokalemia (3.2 mEq/L, 3.5–5.8 mEq/L), elevated alkaline phosphatase activity (402 UI/L; reference range, 46.2–337.2 UI/L) and elevated C-reactive protein levels (210 mg/dL; reference range, 0–35 mg/dL). Computed tomography (CT) was performed to evaluate the anorectal region and detect evidence of metastasis. The mass (14 × 34 mm) was completely blocking the anus (Figure 3B,C). Signs of metastasis were observed on CT. Popliteal, sacral, and iliac lymph nodes were abnormally enlarged. A solid mass was observed in both the liver (23.2 × 16.2 mm) and spleen (16.8 × 10.1 mm). The dog was diagnosed with anorectal cancer. Considering the possibility of metastasis, the owner decided to proceed with salvage surgery to resolve the anal blockage without any further treatment.

Rectal pull-through surgery combined with anal sphincter replacement was performed using a semitendinosus muscle flap (Figure 4). The preoperative preparation and anesthesia were the same as in Case 1. A circumferential incision was made around the anus. Due to the direct invasion of the mass to the external anal sphincter, the anorectum and external anal sphincter were removed altogether. After careful dissection from the surrounding tissue, the affected tissue was retracted caudally and transected, leaving a 1.5 cm cuff of the nondiseased rectal wall. Stay sutures were placed on the rectal wall for better retraction. After covering the anal region with wet gauze, the semitendinosus muscle flap was prepared in the same way as in Case 1, which was through an additional skin incision on the caudal margin of the right hind limb. The flap was passed through the tunnel, created between the two incisions, and the rectum was wrapped around with two bundles of a semitendinosus muscle flap. The muscle bundles were sutured to each other and the pelvic diaphragm was sutured with simple interrupted sutures. Crotch sutures were placed to narrow the rectal lumen. The cut ends of the rectum and skin were sutured together using 4-0 polydioxanone suture. The dog recovered uneventfully and received parenteral nutrition for 2 days. The excised mass and affected anorectal segments were histologically examined. The mass was identified as a highly invasive and scirrhous anal squamous cell carcinoma. This neoplasm originated from the anal mucosa at the anorectal junction. The resection margins of the tumor, skin, and rectum were free of neoplastic cells. The residual feces, impacted in the colon, came out continuously after the anus was opened.

Six days after surgery, the dog achieved complete fecal continence without any complications. At the 3-month postoperative follow-up examination, the anal orifice was approximately 5 mm in diameter at the resting state, and no skin depression around the anus was noticed. The anal reflex was absent. The owner stated that no sign of fecal incontinence or anal inflammation was observed. The dog showed normal voluntary defecation until it died of lung cancer 9 months after surgery.

## 3. Discussion

Causes of fecal incontinence can be categorized into 2 types: reservoir incontinence and sphincter incontinence [1,12,13]. Reservoir incontinence indicates the failure of the large intestine to adapt and contain feces. Animals with reservoir incontinence sense the urge to defecate but are incapable of resisting impulses, leading to frequent defecation of soft, unformed, or liquefied feces. Common causes of reservoir incontinence in dogs include inflammation, surgical resection of the colorectal region, and neoplasia [1,14,15]. Sphincter incontinence indicates the failure of the anal sphincter mechanism to tolerate the propulsive force in the rectum. Affected animals defecate involuntarily without recognition of the passage of feces. Sphincter incontinence can be caused by neurogenic or nonneurogenic disorders. Common causes of neurogenic sphincter incontinence include intervertebral disk disease, cauda equina, neoplasia, and congenital malformations [14,15,16]. Nonneurogenic sphincter incontinence-associated diseases include accidental or iatrogenic injury to the sphincter, perianal disease, rectal prolapse, perineal hernia, and neoplasia [1,3,12].

In Case 1, the external anal sphincter, especially the right side, was severely impaired by surgical site complications and iatrogenic surgical injuries in the anorectal region. The internal anal sphincter might also be damaged due to repetitive infection and surgical manipulation. In addition to muscle tissue, it is highly probable that the surrounding nerves also sustain damage, such as pudendal, rectal, and perineal nerves. The dog involuntarily defecated without recognition, which is indicative of sphincter incontinence. The dog in Case 2 had total removal of the affected anorectum with the surrounding soft tissues, including the external anal sphincter. Both cases in this study can be categorized as sphincter incompetent fecal incontinence. In addition, damage to the external sphincter muscle was not limited to one side of the anus in either dog.

Fecal continence depends on the intricate coordination between neural reflex and muscular activity at the anorectal high-pressure zone [1,12]. The passive (resting) high-pressure zone is created mainly by the internal anal sphincter and subsidiarily by the external anal sphincter and levator ani muscles. The short contraction of the external anal sphincter creates an active high-pressure zone. These two types of high-pressure zones maintain fecal continence.

There are several surgical options for sphincter incontinence in dogs. The adequate surgical technique is determined according to the cause of fecal incontinence and the purpose of surgical treatment. The implantation of polyester-impregnated silicone elastomer or fascial sling, increasing the passive high-pressure zone, can be used to replace or augment the anal sphincter function. These methods have disadvantages in that artificial implants are susceptible to infection, especially in the anorectal area, and loosening may occur intra- or postoperatively. Additionally, persistent tenesmus can occur after surgery [5]. To reinforce the active high-pressure zone, applications of the functional structure have been reported, such as semitendinosus, sartorius, levator ani, and latissimus dorsi muscle flap transposition. Coaptation to the pudendal nerve or end-to-side pudendal nerve anastomosis may be beneficial in gaining more control over the active contraction of the muscle flap [9,10].

Muscles suitable for transfer without microanastomosis are those whose origin can be transposed to the defect site without neurovascular compromise and those that are large enough to cover the defect area. Accessibility and vascular pattern are also important factors of the muscle flap. There are a limited number of muscles around the anus satisfying these criteria. In cases of total defects of the external anal sphincter, such as the present cases, no muscle transposition method has been reported in dogs, except artificial implants or free tissue flaps. In the present study, the semitendinosus muscle was used to reinforce the entire external anal sphincter. In particular, in Case 2, the muscle flap was intended to replace the whole striated musculature involved in fecal continence. By bifurcating the end of the semitendinosus, the anorectum could be completely encircled by two bundles of this muscle. This allows total replacement of the external anal sphincter with only one muscle flap, without using additional muscle to reach the uncovered defect or wrapping a muscle flap around the anorectum 360 degrees (Figure 5). Additionally, it enables a shorter semitendinosus flap length compared to conventional techniques, in which the semitendinosus muscle is transected close to its insertion on the tibia. Usually, only a distal part of the transposed muscle flap (with the least blood supply) is used for anal sphincter reconstruction. Therefore, long pedicled muscle flaps could lead to partial fibrosis, malfunctioning, and avascular necrosis [17]. Approximately half to two-thirds of the length of the semitendinosus muscle was used in this study. The semitendinosus muscle is supplied by two dominant vascular pedicles from the proximal (branch of the caudal gluteal artery) to the distal ends (branch of the distal caudal femoral artery) [18,19]. These two pedicles are connected through microanastomosis in the middle of the muscle belly [18]. By shortening the length of the transposed muscle, this technique minimized the risk of vascular compromise without overtensioning the flap and allowed the muscle portion, with the optimal blood supply, to be placed around the anorectum, leading to superior contractile function.

The major limitation of this method was that the semitendinosus muscle was not innervated by the pudendal nerve. Therefore, the transposed flap did not coordinate with other components controlling fecal continence, and the flap alone was insufficient for restoring neurogenic control of the external anal sphincter. However, there have been studies documenting the establishment of neuronal continuity (reinnervation) from the recipient bed to the transposed muscle flap [20,21,22]. These studies showed that the ingrowth of the pudendal nerve over a gap between the semitendinosus muscle flap and surrounding muscles could lead to coordinated movements of these muscles and reinforce the physiological function of the remaining external anal sphincter. Another limitation is that splitting up in the middle of the semitendinosus muscle could disturb the vascularity of the flap and reduce the thickness of the transposed muscle, which will decrease the functionality of the flap. The ideal muscle flap for the anal sphincter is a skeletal muscle with sufficient blood supply, innervated by the pudendal nerve. Therefore, the bifurcated semitendinosus muscle flap might not be an ideal surgical method satisfying all conditions of optimality.

The two dogs in this study regained decent levels of fecal continence after surgery. The dog in Case 2 had complete control over defecation. The actual function of the transposed semitendinosus muscle flap could not be verified. However, considering the outcomes and follow-up periods, those flaps might function as a supportive mechanism by increasing the passive high-pressure zone. While no evidence of functional recovery was identified in this report, it is worth considering that functional restoration by reinnervation may occur following an extended recovery period.

## 4. Conclusions

These case reports present a modified semitendinosus muscle flap transposition for the augmentation or replacement of the external anal sphincter function in dogs. This technique showed acceptable outcomes with improved fecal continence in two dogs. Considering that the surgical procedure is relatively simple and the entire external anal sphincter can be augmented, bifurcated semitendinosus flap transposition could be one of the surgical treatment of anal sphincter incontinence in dogs. However, the case number is too small to draw definite conclusions. In addition, this technique has an obvious limitation because nervous coordination with other fecal continence controlling components was not established. Further studies are needed to objectively evaluate the clinical consistency and overcome the limitations of the bifurcated semitendinosus muscle flap transposition.

## Figures and Tables

**Figure 1 vetsci-10-00150-f001:**
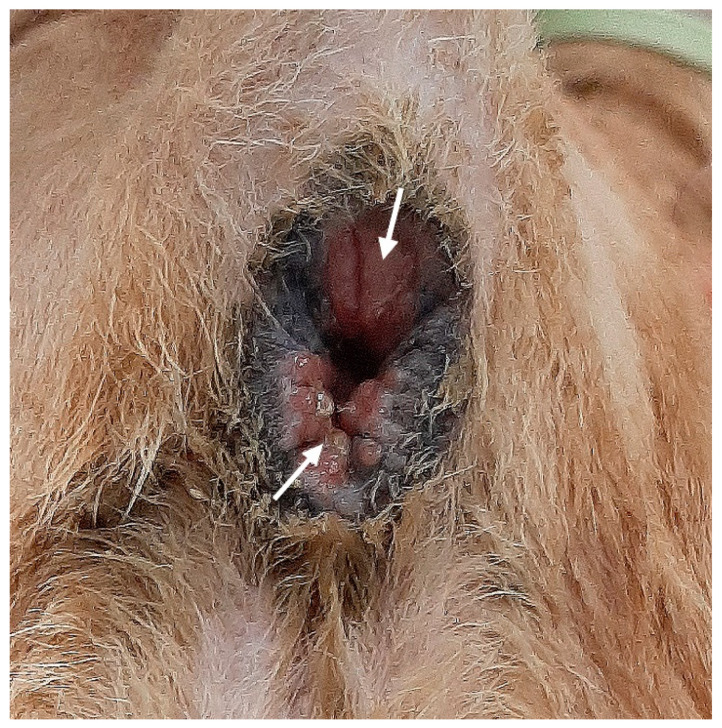
Anal area with sphincter incompetence. The anal orifice is abnormally widened with the rectal mucosa (white arrows) exposed outside.

**Figure 2 vetsci-10-00150-f002:**
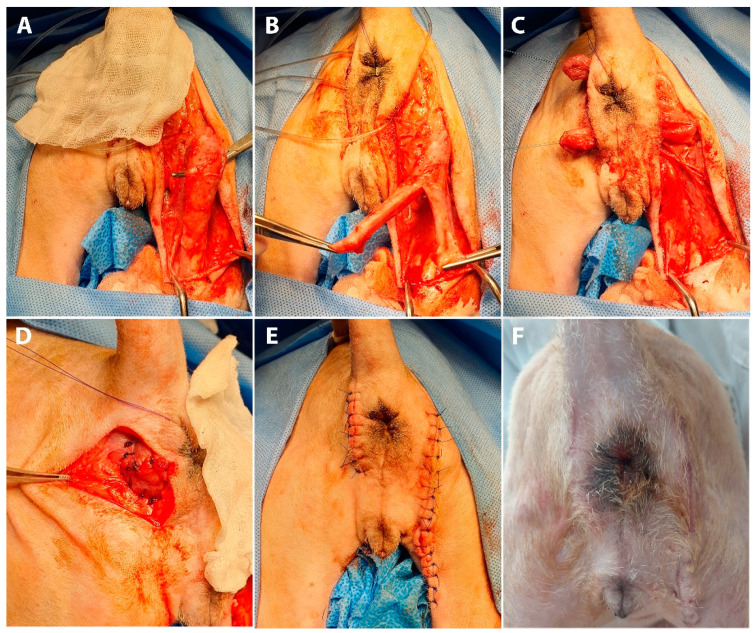
Intraoperative (**A**–**E**) and postoperative (**F**) views of bifurcated semitendinosus muscle flap transposition in a dog with anal sphincter incompetence. (**A**) The semitendinosus is carefully dissected to minimize damage to the blood and nerve supply to the proximal part. (**B**) After the semitendinosus muscle is transected at the level between the middle and distal one-third, the distal half is divided into two bundles by longitudinally cutting the middle part of the muscle body. (**C**) Two bundles are passed through the tunnels created at the dorsal and ventral sides of the anus. (**D**) The semitendinosus muscle bundles are sutured to the adjacent rectal muscle and the pelvic diaphragm, and crotch sutures are placed between two muscle bundle ends to narrow the anorectal lumen. (**E**) After surgery, the anus was successfully narrowed without exposed mucosa. (**F**) At the three-month postoperative follow-up, the morphology of the anal orifice is normal.

**Figure 3 vetsci-10-00150-f003:**
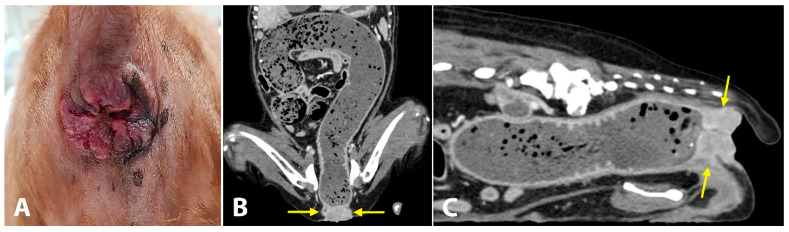
Photograph (**A**) and computed tomography (CT) images (**B**,**C**) of anal mass in a dog. (**A**) Solid mass with an ulcerative and hyperplastic surface at the anal region. (**B**,**C**) CT shows an anal mass (yellow arrows), completely blocking the passage of feces. The entire colon is severely dilated.

**Figure 4 vetsci-10-00150-f004:**
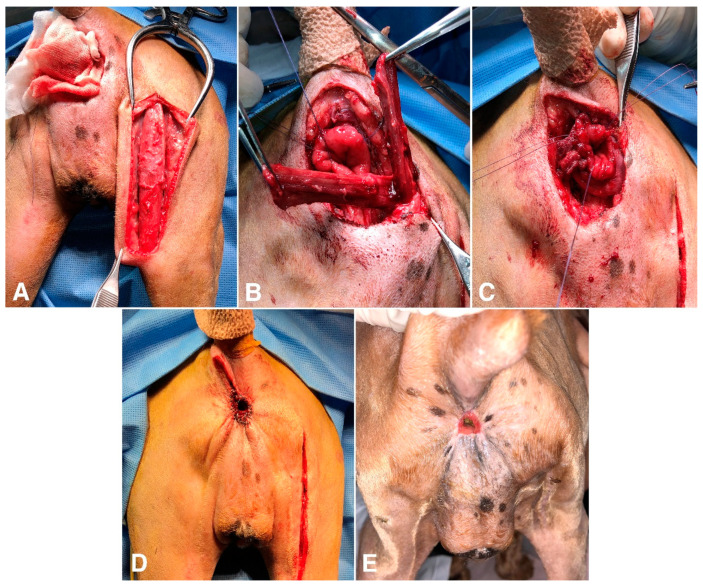
Intraoperative (**A**–**D**) and postoperative (**E**) views of anorectal resection with a bifurcated semitendinosus muscle flap in a dog with an anal tumor. (**A**) After the anal mass was resected en bloc with clear margins, the semitendinosus muscle was carefully dissected from the surrounding tissue. (**B**) The semitendinosus flap was transposed to the anal region through the subcutaneous tunnel, and the distal end of the flap was bifurcated into two bundles. (**C**) The two bundles were wrapped around the anus and suture the ends of the bundles with each other to narrow the intestinal lumen. (**D**) Intestinal transected edges were sutured to the skin, leaving a small passage for feces. (**E**) At the postoperative three-month follow-up, the reconstructed anus maintained its narrowed orifice diameter (5 mm), and the dog showed a normal defecation pattern.

**Figure 5 vetsci-10-00150-f005:**
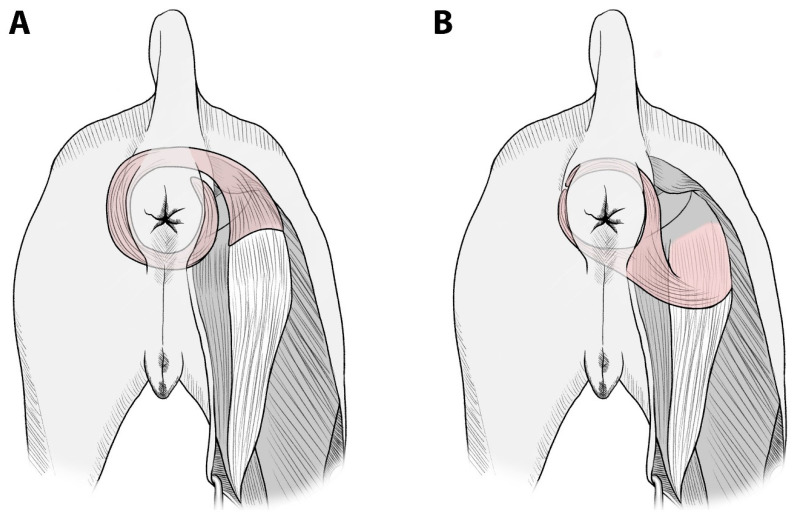
Illustration of the two possible options of semitendinosus muscle transposition for total replacement of the external anal sphincter in a dog. (**A**) To completely encircle the anus, a long semitendinosus muscle is needed. (**B**) By bifurcating the distal end of the semitendinosus muscle, the required flap length can be substantially reduced, leading to less impairment of the integrity of the vasculature and nerve supply to the proximal part of the semitendinosus muscle.

## Data Availability

The data presented in the study are available on request from the first and corresponding author. The data are not publicly available due to ethical and privacy concerns.

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
