# Peer review of "Application of Bifurcated Semitendinosus Muscle Transposition for Treatment of Fecal Incontinence in Two Dogs"

_vetsci, 2023, doi:10.3390/vetsci10020150_

Round 1

Reviewer 1 Report

well written paper of practical interest to soft tissue surgeons; for specific comments please see attached Word file

Author Response

January 10, 2023

Resubmission of manuscript Application of Bifurcated Semitendinosus Muscle Flap for Fecal Incontinence in Two Dogs

Thank you for the opportunity to revise our manuscript. We appreciate the careful review and constructive suggestions. We believe the manuscript is substantially improved after making the suggested edits.

Following this letter are the editor and reviewer’s comments with our responses, including how and where the text was modified. Changes made in the manuscript are highlighted in red color. The revision has been developed in consultation with all coauthors, and each author has approved this revision’s final form.

Sincerely,

Mu Young Kim

Comment 1: Although the title is appropriate, the use of a bifurcated semitendinosus muscle flap should not be called “a novel method” in the text. The use of this muscle belly transfer, encircling partially or completely the anus has been described before (1991 and 2003) in 2 and in 1 dog, respectively; (the citations are correctly included). The technique may better be called a modification of the flap by splitting the ends, but nothing more…in calling it a modification instead of a novel technique, the paper does not loose quality or interest.

Response: We agree with the review. We replaced “a novel method” with “modified”.

  • Line: 15, 29, 72,320

Comment 2: For the sake of clarity and for avoiding any possible misunderstanding, I suggest to more clearly stating that this flap serves as augmentation, that it improves mechanically (by compression) the largely uncontrolled passage of feces and that it cannot restore functional, neurogenic control of the external anal sphincter mechanism. (Although on lines 279 ff the lack of pudendal nerve innervation is described as shortcoming, yet the concept of only-augmentation should be stated clearer).

Response: We agree with the review. We revised our manuscript clearly stating that this muscle flap served as augmentation.

  • Line: 16, 37, 298-300, 314-318

Comment 3: Also, in the Discussion, in case 1, the previous iatrogenic damage to one or both caudal rectal nerves should be discussed (a not uncommon complication of excessive perianal fistula excision) as this is different from case 2, where the entire anal-anorectal compartment including its innervation was excised and a routine pull-through procedure performed.

Response: We agree with the review. However, we couldn’t specify which nerved was damaged. So, we added the sentence about the possibility of nerve damage around the surgical area.

  • Line: 238-239

Comment 4: Given the histologically confirmed high malignancy degree of the anorectal carcinoma, the peri-anal skin and sub-lumbar lymph node involvement, and the distant organ metastases, the 9-month survival is surprising, and, per se, a palliative treatment success; if there was any chemotherapy protocol used, please describe.   

Response: We didn’t perform chemotherapy in this case. The surgery was performed for palliative purposes.

Comment 5: Line 17 change sentence “novel”

Response: We agree with the review. We changed this part.

  • Line: 29

Comment 6: In keywords: add “dog”

Response: We agree with the review. We added the “dog” in keywords.

  • Line: 39

Comment 7: “vagus ”: it is unnecessary to list the vagus nerve separately as the pelvic nerves (which you correctly mention) are the parasympathetic innervation of the rectum; “The vagal nerve controls the right colon and the pelvic nerve permeates the left colon and rectum via the rectal branches of the pelvic plexus” . Also, I would specify ….”and the motoric pudendal nerve” – as opposed to the autonomous hypogastric and pelvic nerves.

Response: We agree with the review. We changed the sentence as the reviewer commented.

  • Line: 47-48

Comment 8: Line 49: this electrostimulation was in an experimental model; please state this.

Response: We agree with the review. We stated that this electrostimulation was in an experimental model.

  • Line: 61

Comment 9: Line 60: without “devising” a method, it is perfectly acceptable to report an additional (4th) case of flap transposition, this time a modification of the method of the 3 previously reported cases.

Response: We agree with the review. We changed it.

  • Line: 74

Comment 10: Line 78: widely open

Response: We agree with the review. We changed it.

  • Line: 91

Comment 11: Line 80: what do you mean by sphincter decreased by rectal palpation ? the tone of the sphincter or the muscular volume (which latter is difficult to assess by rectal palpation); was there perhaps only unilateral denervation ? well, this remains speculative, although on line 227 you state it was bilateral…..

Response: We agree with the review. We decided to remove this sentence to prevent confusion. The actual intention of that sentence was to show that the volume of soft tissue was significantly reduced.

  • Line: 92-94

Comment 12: Line 137:….occasionally continued to leak…

Response: We agree with the review. We changed it.

  • Line: 151

Comment 13: Line 235: “optimal” there is no optimal surgical solution; recommend saying “adequate or best solution”

Response: We agree with the review. We changed it.

  • Line: 252

Comment 14: Line 254: add the ….whole “striated” musculature…. (that is the external sphincter); if you had excised also the (whole) internal sphincter, the outcome would certainly have been complete fecal incontinence.

Response: We agree with the review. We added “striated”.

  • Line: 271

Comment 15: Line 299 “good outcomes” this is relative and remains disputable; perhaps you should say “acceptable outcomes” (which does not diminish the positive message the report is transmitting). 

Response: We agree with the review. We replaced “good” with “acceptable”.

  • Line: 322

Reviewer 2 Report

Many thanks to the authors for a concise and very interesting manuscript describing a new way to use the semitendinosus muscle flap for fecal incontinence in two dogs. The authors describe in detail their technique, explaining why they did what they did, and, furthermore, they are aware of the limitations of their report. This technique will surely help many an animal in the near future. I have just a few comments, in an attempt simply to further improve the manuscript, and they can all be found in its text.

Author Response

January 10, 2023

Resubmission of manuscript Application of Bifurcated Semitendinosus Muscle Flap for Fecal Incontinence in Two Dogs

Thank you for the opportunity to revise our manuscript. We appreciate the careful review and constructive suggestions. We believe the manuscript is substantially improved after making the suggested edits.

Following this letter are the editor and reviewer’s comments with our responses, including how and where the text was modified. Changes made in the manuscript are highlighted in red color. The revision has been developed in consultation with all coauthors, and each author has approved this revision’s final form.

Sincerely, Mu Young Kim

Comment 1: Consider adding "fecal incontinence" and  "(semitendinosus) muscle (flap)"

Response: We agree with the review. We added these two into keywords.

  • Line: 39

Comment 2: I suppose in size and not in strength; however, consider being more specific...

Response: We agree with the review. We decided to remove this sentence to prevent confusion. The actual intention of that sentence was to show that the volume of soft tissue was significantly reduced.

  • Line: 92-94

Comment 3: Which was the reason for using maropitant, an antiemetic, as a premedicant in this case? I certainly approve the use of famotidine in order to reduce the impact of a potential gastroesophageal reflux episode, but why use an antiemetic also? After all, as far as we know, butorphanol seems also to have an antiemetic effect, at least when a2 agonists are used...

Response: We agree with the review. In our cline, maropitant was used as premedication to prevent intraoperative vomiting and for supplementary analgesic effect as part of a multimodal analgesic plan.

  • Line: 238-239

Comment 4: The analgesic effect of butorphanol is largely dismissed by most anaesthetists. Although better than nothing, butorphanol is used as a pre-anaesthetic agent rather to augment the effect of the sedative/s used and not as an analgesic. Any post-operative analgesia your dogs had should be attributed to carprofen and not to butorphanol. However, if you had no access to opioids other (morphine, pethidine, faintanyl) than butorphanol, please indicate it.

Response: We agree with the review. We understood the limitation of butorphanol as an analgesic, but at the time point of this case, we were not available of other opioids. We added the sentence, explaining about it.

  • Line: 140-141

Comment 5: Request for all the grammatical errors and minor errors.

Response: We agree with the review. We revised every error the reviewer pointed out.
